# Measuring Maritime Paint Thickness under Water Using THz Cross-Correlation Spectroscopy

Johan Østergaard Knarreborg [1,†], Jonathan Hjortshøj-Nielsen [1,†], Bjørn Hübschmann Mølvig [2], Thorsten Bæk [2], Peter Uhd Jepsen [2] and Simon Jappe Lange [2,*]

1    DTU Energy, Technical University of Denmark, 2800 Kgs. Lyngby, Denmark
2    DTU Electro, Technical University of Denmark, 2800 Kgs. Lyngby, Denmark
*    Correspondence: slla@dtu.dk; Tel.: +45-22-98-77-99
†    These authors contributed equally to this work.

**Abstract:** The shipping industry is a major contributor to global greenhouse gas (GHG) emissions, which is why it is important to optimize every aspect of the efficiency of ocean-going vessels. This includes the antifouling paint that ensures hydrodynamic efficiency. Measuring the thickness of the antifouling on top of all other paint layers using THz cross-correlation spectroscopy (THz-CCS) underwater could enable vessel operators to monitor the state of the paint on ship hulls and plan any vessel's sailing route and maintenance optimally. However, due to the high absorption of water in the THz domain, measuring through any significant amount of water is impossible, making a water removal method necessary. This study shows how a THz-CCS system can be packaged for underwater measurements using a molded silicone contact seal. In combination with a spectroscopic model for data treatment, the thickness of a single paint layer is retrieved underwater. This paves the way for a more advanced system capable of measuring multilayer maritime paint underwater, which will enable shipping companies to continuously monitor the paint layers' thickness.

**Keywords:** terahertz cross-correlation spectroscopy; NDT; antifouling paint; stochastic fitting algorithm; time-of-flight analysis; thickness

## 1. Introduction

The shipping industry accounts for 2.7–2.9% [1–3] of all anthropogenic greenhouse gas emissions (GHG), equivalent to 1076 million tonnes of $CO_2e$ each year [1,3]. As a result, the industry is subject to increasing regulations and pressure to reduce GHG emissions in the interest of reducing its global carbon footprint [4]. Major organizations, such as the International Maritime Organisation, are committing to a reduction of GHG of 50% by 2050 compared to 2008 [5], fueling the demand for technical solutions that can help increase the fuel efficiency of large vessels such as container ships.

A vital component to ensure the efficiency increase is the ability of vessel operators to constantly monitor the quality and condition of the vessel paint during docking sessions and active operation. The underlying reason is that the paint highly influences the hydrodynamic properties of the vessel, most prominently by minimizing the drag through the prevention of fouling [6]. Fouling is the accumulation of organic material such as biofilm, algae, and barnacles [7]. Fouling on vessel surfaces greatly affects the overall drag, increasing fuel consumption by up to 30% in extreme cases [8,9]. The practical way to prevent fouling is to apply antifouling paint on vessels as the outermost paint layer [7]. This paint is applied on top of anticorrosive and binding layers. The most common antifouling paint is by design "self-polishing" [10], which means that the paint slowly deteriorates to expose new clean antifouling paint, thus removing any fouling accumulation. In cases where the antifouling paint wear is higher than anticipated, or the paint layer is thinner than estimated, the self-polishing process leaves the vessel partly unprotected [8]. Consequently, ships are sandblasted and repainted upon dry-dock sessions, but this is a costly

process due to labor, paint, rerouting the vessel, and the lost operating time [11], which encourages the shipping companies to extend the periods between docking sessions as long as possible. In turn, it has been estimated that 10% of the fuel consumption of ships stems from inefficient surface performance of the hull [12]. Since most antifouling paint is self-polishing, its condition can at any point during vessel operation be well-described by its remaining paint thickness. Shipping companies can consequently use continuous paint thickness monitoring to optimize the fuel efficiency of their vessels, leading to a reduction in GHG emissions.

On top of the GHG emission increase, biofouling on ship hulls leads to invasions of nonindigenous marine species in foreign waters. This can lead to the introduction of new infectious diseases and threaten the biodiversity of freshwater, estuarine, and marine ecosystems [13]. By knowing the exact remaining thickness of antifouling paint, it is possible to establish procedures that can effectively mitigate the introduction of invasive species by reducing the amount of biomaterial attached to container ship hulls [14].

Currently, there are several methods used to measure the full-stack thickness of ship paint including all layers given as one number. A fast, reliable, and accurate way of doing this is a device utilizing the Hall effect and eddy currents to measure a wide variety of dry paint types [14,15]. Ultrasound has also proved to be able to accurately and reliably measure the thickness of ship paint [16]. In the case of wet paint, which is relevant upon paint application during a dry-dock session, another used method is simple pins of different heights pushed through the paint, called a "Wet Film Gauge". The number of pins in contact with the paint corresponds to the absolute thickness of the paint [17]. One drawback of this method is that the thickness of the wet paint shrinks, as the paint dries. Therefore, a wet paint thickness measurement can be unreliable [18]. All these methods cannot measure individual layers in a full stack, which is relevant since ship paint usually consists of three parts: an anticorrosive layer, a binding layer, and the antifouling paint as the outermost layer [19]. Current technology, therefore, fails to measure the antifouling layer separately.

It has been demonstrated that terahertz (THz) lightwaves can be used to detect and quantify individual layers in a multilayer paint stack [20,21]. This is typically done using either THz time-domain spectroscopy (THz-TDS) or THz frequency domain spectroscopy (THz-FDS), whilst the more recent modality, THz ellipsometry, is also gaining interest [22]. The common trait for all THz-based measurement techniques is that the broader the frequency spectrum, the more precise thickness measurement one can obtain. In addition, the thinnest measurable layer thickness is normally determined by the highest available THz frequency in the spectrum. However, THz technologies typically have a decreasing output power with increasing THz frequency. Higher THz frequencies additionally tend to have higher absorption coefficients in most materials, which further exaggerates the challenge of being able to measure layer thicknesses with high THz frequencies—particularly in noncontrolled environments in the field. Current commercially available THz technologies address these issues by employing very large bandwidth systems at the expense of price and footprint. This in turn leads to a very limited field use of THz technologies in general [23].

The issue of THz absorption in materials is particularly problematic when it comes to water. At 1 THz, the absorption coefficient is above $200 \, \mathrm{cm}^{-1}$ [24], leading to a 98% signal loss through a $100 \, \mu\mathrm{m}$ water layer as the beam would have to travel through the water layer twice. In practice, this means that measuring paint layer thicknesses through water is not feasible. In the case of container ships, however, the relevant part of the hull for paint monitoring resides below the water line [25,26].

In this work, we show that it is possible to measure a single paint layer underwater using a THz cross-correlation spectroscopy (THz-CCS) system. The THz-CCS system is sufficiently small and robust to be envisioned for field applications. In addition, the system is designed to operate mainly with low THz frequencies to minimize absorption loss in materials that are inspected.

Figure 1 shows a conceptual implementation. The THz-CCS system sits in a dry environment and is coupled to a water-proof measuring head that can be operated below the water line. The head mainly contains a THz transmitter (Tx) and THz receiver (Rx), as well as a contact patch to the ship hull. The contact patch is pressed onto the ship hull to create a watertight seal. The THz radiation passes through the contact patch, interacts with the ship paint, and information is reflected back to the receiver. The primary components, which are developed and demonstrated in this work, are the underwater contact patch and data analysis algorithm for extracting paint thickness through the patch underwater.

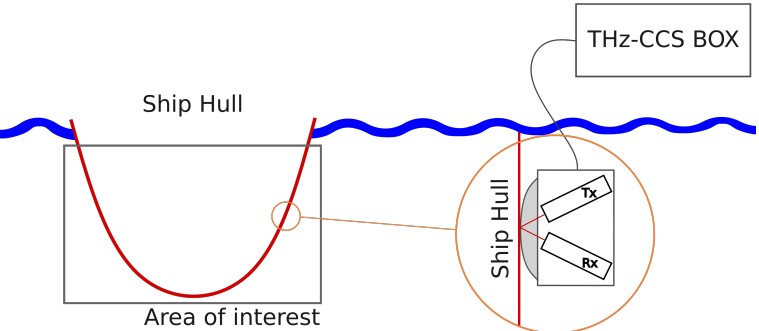

**Figure 1.** Outline of a potential solution, showing the area of interest on the hull, the underwater part of the prototype containing the THz antennae with a silicone contact patch, and the box containing other necessary components of the THz-CCS system above water.

## 2. Materials and Methods

### 2.1. THz-CCS System

The THz-CCS system used in this paper was developed at the Department of Electrical and Photonics Engineering at the Technical University of Denmark. The THz-CCS system was proven useful in several settings including measuring variations in paper thicknesses, with thicknesses on the same order of magnitude as ship paint [27].

Standard THz-TDS systems usually use a single pulse from a femtosecond laser to generate a signal. To reduce the size of the device and enable the use of fiber optic cables throughout the whole system for increased robustness, the THz-CCS system replaces the femtosecond laser with a broadband CW output from a C-band amplified spontaneous emission (ASE) light source. The delay stage of the THz-CCS system does not rely on free-space propagation, as in most THz spectroscopy systems. In the THz-CCS system, light is restricted in fiber optic cables wrapped around piezoelectric crystals than can stretch the cables, resulting in a delay of the light as in [28]. By utilizing a fiber-coupled light source, the system is more resilient to vibrations, which inevitably will affect the system outside of a laboratory. The full THz-CCS system, other than the antennae, is enclosed in a transportable 25 cm × 25 cm × 10 cm box with an IP66 rating [29], and the antennae, seen in Figure 2A, are connected by fiber optic cables. This makes it possible to keep the box above water and only submerge the antennas; hence, only the antennas have to be waterproof. This concept can be seen in Figure 1.

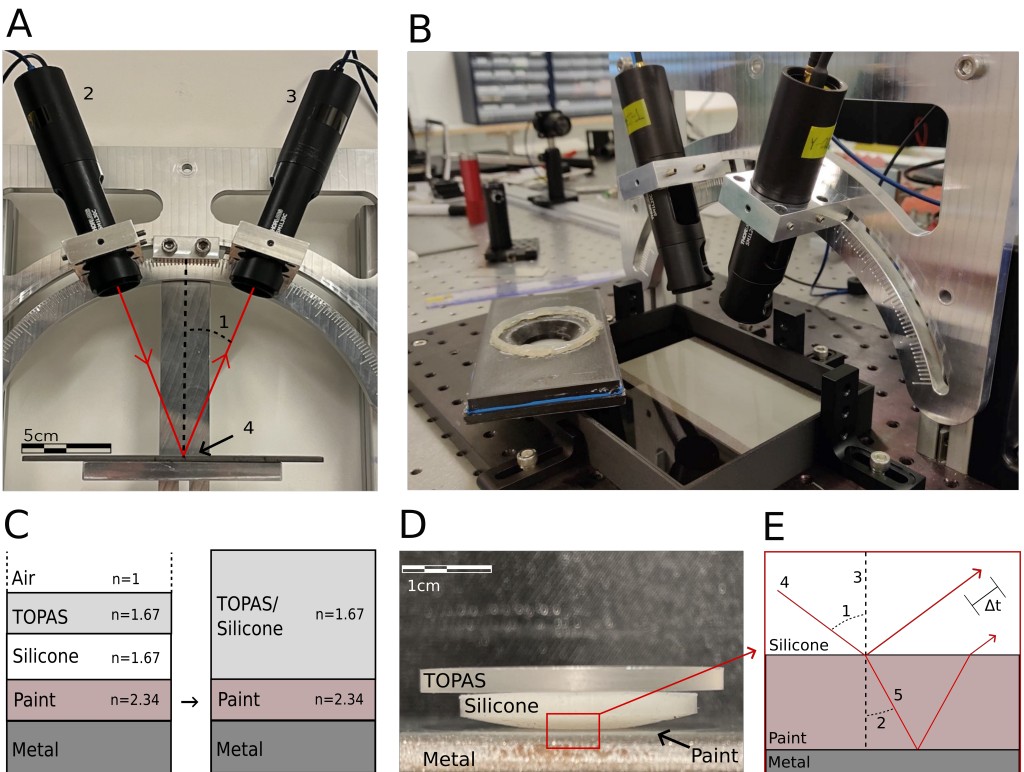

**Figure 2.** (**A**) Traditional reflection setup with the paint placed at the focal point (4) of the transmitting (2) and receiving (3) antennae. The antennae are set at an angle of incidence of 20° (1) and use the s-polarization. (**B**) The silicone and TOPAS plastic contact patch on top of the paint sample. In the measurements, the contact patch was encased in a 3D-printed device, which enabled pressure to be applied to the silicone. (**C**) Zoomed-in version of the illustration on how the thickness can be derived from the difference in arrival time ($\Delta t$) of the two reflections, described by Equation (1). (**D**) The TOPAS and silicone layers on the paint, without the 3D-printed holder. (**E**) A simplification of the physical model was made to reduce the complexity.

## 2.2. Reflection Setup

The THs-CCS system is operated in a reflection setup with a nonzero angle of incidence, as seen in Figure 2A. With this setup, the THz beam is directed at an angle (**1**) towards the sample located at the focal point (**4**). Due to the nonzero angle of incidence and the different refractive indices of the media at the interface, light is refracted onto the paint sample. This is illustrated in Figure 2E.

A simple way to calculate the thickness of a medium is by comparing the time difference between the reflection from the paint and the metal underneath. The time difference in reflections can be seen as $\Delta t$ in Figure 2E and in the time-domain in Figure 4A. This leads to a description of the paint thickness ($T$) as:

$$T = \cos\left(\sin^{-1}\left(\frac{n_1}{n_2}\sin(\theta_1)\right)\right)\frac{c\Delta t}{2n_2} \tag{1}$$

with $n_1$ and $n_2$ the refractive index of the material above the paint and the paint, respectively. $c$ is the speed of light, and $\theta_1$ is the angle of incidence.

## 2.3. Spectroscopic Model

As discussed in Section 2.2, reflections from different interfaces are separated in the time domain as seen in Figure 4A. With a clear separation of the reflection peaks in the time domain, the thickness can be found by a simple calculation through Equation (1). It should be noted that the refractive indices of the media need to be known to use this method. However, when the media thicknesses become smaller, the different reflection

peaks overlap significantly, and the location of the peaks in the time domain can no longer be reliably determined. This problem can be solved by using a more advanced spectroscopic model, where both an unknown thickness and refractive index can be determined. This procedure is now introduced.

The transfer function of a three-layer system is considered. When a plane wave $E_0$ coming from medium 1 is incident on medium 2, the total reflected field is the sum of the initial reflection and all internal reflections between medium 2 and 3, i.e.,

$$\tilde{E}_r(\omega) = \tilde{E}_0\left(r_{12} + t_{12}e^{-i\phi}r_{23}e^{i\phi}t_{21} + ...\right) \tag{2}$$

$$= \tilde{E}_0\left(r_{12} + t_{12}t_{21}r_{23}e^{-2i\phi}\sum_{n=0}^{\infty}\left(e^{-2i\phi}r_{23}r_{21}\right)^n\right)$$

where $r_{ij}$ are the appropriate Fresnel reflection coefficients and $\phi$ is the acquired phase upon one traversal of medium 2 given by

$$\phi = \frac{\omega \cdot n(\omega) \cdot d}{\cos\theta \cdot c}, \tag{3}$$

where $\omega$ is the angular frequency, $d$ is the layer thickness, and $\theta$ is the angle of incidence. Simplifying Equation (2), we find that

$$\tilde{E}_r(\omega) = \tilde{E}_0(\omega)\frac{r_{12} + r_{23}e^{-2i\phi}}{1 + r_{12}r_{23}e^{-2i\phi}}. \tag{4}$$

If an additional medium is present behind medium 3, the total reflected field from medium 3 to medium 2 is calculated by applying Equation (4) again to determine an effective expression for $r_{23}$ labeled $H_3(\omega, d, n)$. This procedure is recursively followed until all layers are accounted for [30].

With a reference measurement $\tilde{E}_{ref}(\omega)$ from a highly reflective metal plate with Fresnel coefficient $r_m$

$$\tilde{E}_{ref}(\omega) = \tilde{E}_0(\omega)r_m \tag{5}$$

and a model for the refractive index $n$, the thickness $d$, and refractive index parameters can be determined for each layer by minimizing

$$L(d_1, n_1, ..., d_n, n_n) = \left|\frac{\tilde{E}_{sample}}{\tilde{E}_{ref}} - \frac{r_{12} + H_3e^{-2i\phi}}{r_m\left(1 + r_{12}H_3e^{-2i\phi}\right)}\right|^2. \tag{6}$$

The silicone layer is modeled as having a constant refractive index $n$ and the paint layer is modeled as a Debye medium with a permittivity function given by

$$\epsilon(\omega, \epsilon_\infty, \epsilon_s, \tau) = \epsilon_\infty + \frac{\epsilon_s - \epsilon_\infty}{1 - i\omega\tau}, \tag{7}$$

where $\epsilon_\infty$ is the high-frequency limit of the permittivity, $\epsilon_s$ is the static permittivity, and $\tau$ is a material-dependent relaxation time.

Even for a few layers, Equation (6) quickly becomes highly nonconvex, and therefore a local minimization algorithm is inappropriate for the parameter extraction. Hence, the fitting is done with the stochastic optimization algorithm differential evolution [31], which has previously been shown to perform well for layer thickness retrieval based on THz measurements [30]. To quantify the goodness of fit, the coefficient of determination $R^2$ is used. If $y$ is the measured pulse and $\hat{y}$ is the best fit, $R^2$ is defined as

$$R^2 = 1 - \frac{\text{Var}(y - \hat{y})}{\text{Var}(y)}, \tag{8}$$

such that a perfect fit corresponds to $R^2 = 1$.

From Snell's law and Fresnel's equations, it is evident that neither reflection nor refraction takes place at the interface of two materials with the same refractive index. At the interface between TOPAS and silicone, a nonsignificant angle of refraction and a nonsignificant reflection is expected due to the two materials having a near identical refractive index. This makes it possible to use a simplified physical model, as seen in Figure 2E. This was done to make the calculations for the algorithm simpler and more reliable. Additionally, it is worth noting that the runtime of the fitting algorithm is not important, as only point measurements are taken, and an immediate result is not required.

### 2.4. Reference Measurement

The paint sample shown in Figure 3A was a 14 cm by 7 cm brushed steel plate with one layer of Jotacote Uni N10 Alu paint applied. The sample was provided by the company JOTUN. The expected dry film thickness (DFT) that was calculated by the amount of paint applied was 200 µm.

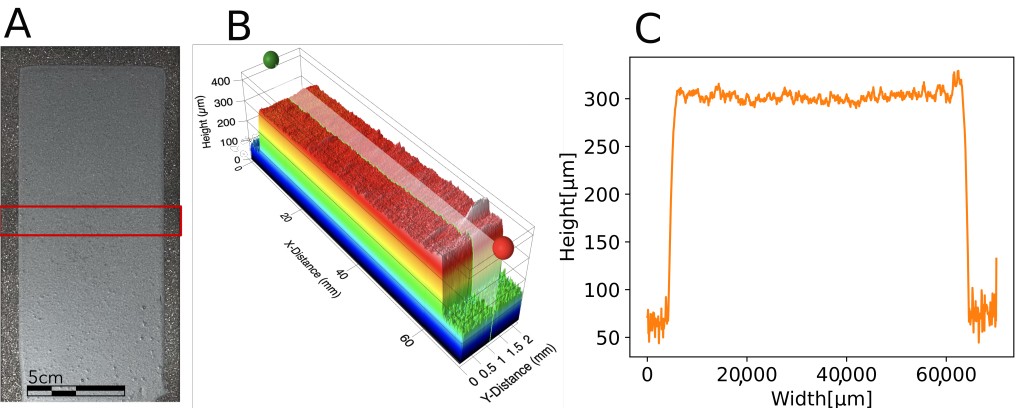

**Figure 3.** Determining the thickness of the given paint sample: (**A**) The sample given by Jotun. The red square indicates the area of the 3D scan. (**B**) Three-dimensional scan of the paint. (**C**) Height along the line indicated in (**B**). An average thickness can be derived from the difference in the average height between the metal and the top of the paint. For the paint sample, the average paint thickness was 230.0 µm.

To get an accurate reference measurement for the correct dry film thickness (DFT), we performed a surface height and roughness 3D scan using an Alicona focus-variation microscope. A magnified 3D image of a scan across the sample is visible in Figure 3B and the average surface height along the line in Figure 3B was 230.0 µm. The average surface height standard deviation, i.e., the roughness, was 9.4 µm. Therefore, for a practical field implementation, the layer thickness accuracy needs only to be within 9.4 µm.

### 2.5. Silicone Contact Patch

The high absorption coefficient of water in the THz frequency range makes measurements through any significant amount of water [24] almost impossible. A rounded silicone patch was made to remove all water in the THz beam path, illustrated in Figure 2D. The design philosophy of the contact patch is to have a parabolic end that can push water away and avoid trapping water under the contact patch. A silicone with hardness "Shore A15" was chosen due to it being the softest silicone available, which should be a good candidate to fully make water-tight contact when pressure is applied. At the initial contact between the silicone and the paint, only the center part of the parabolic silicone end touched the paint. When enough pressure was applied, all of the parabolic silicone end touched the paint, and water was pushed away, which resulted in a contact surface large enough to contain the entire THz beam, where full contact was made. Additionally, there was no indication of any loss of contact due to the central part of the contact patch bending away

from the paint. Due to the high Young's modulus of maritime paint relative to silicone [32], it could be assumed that the deformation of the paint was negligible.

The molding of the silicone was done with 3D-printed molds printed with PLA plastic using the Ultimaker 3 3D printer. Due to its low absorption coefficient at THz frequencies, a TOPAS wafer was used to apply pressure to the silicone patch [33].

### 2.6. Underwater Measurements

To simulate an underwater setting, setup (A) in Figure 2 was used with the paint sample fully submerged in water illustrated in Figure 2B. The silicone contact patch and TOPAS polymer were integrated into a 3D-printed holder that made it possible to apply pressure to the silicone and thereby remove water in the path of the THz beam. It should be noted that this experiment only tested the ability to remove water in the path of the THz beam and not the water resistance of a fully integrated device.

## 3. Results and Discussion

### 3.1. Dry Measurement of Paint Sample without Contact Patch

As a first step, we used the THz-CCS system to measure the paint thickness in a dry environment without the contact patch. This was done using the reflection setup shown in Figure 2A with an incident angle of 20°. The resulting sample time trace is shown in Figure 4A. It is clear that two reflections were separated by 3.9 ps. Applying Equation (1) and using tabulated values n ∈ [1.7–2] for the refractive index of the paint [20], the calculated thickness was in the range 273 µm–319 µm, which was far from the actual thickness. This was assumed to be because oscillations following a reflection overlap and the assumed refractive index were not exactly those of the sample. Fitting Equation (6) on a three-layer model with constant indices of refraction instead, the best fit shown in Figure 4C, with a coefficient of determination of $R^2 = 0.960$, determined the paint thickness to be 235 µm and the refractive index to be 2.34. The measured thickness deviated by 5 µm from the Alicona 3D scan, which was within the mean surface height deviation. Therefore, we concluded that the two measurements were in agreement. When using the refractive index of 2.34 of paint from the best fit together with Equation (1), it resulted in a calculated thickness of 234.6 µm. This result was also in agreement with the Alicona 3D scan. The difference in calculated thickness between Equation (6) and the best fit was only 2.2%. This high degree of agreement between the methods suggested that the THz-CCS device could be used to extract the thicknesses and refractive indices of multilayer films correctly.

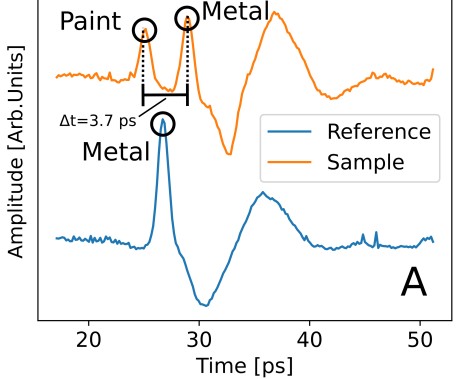
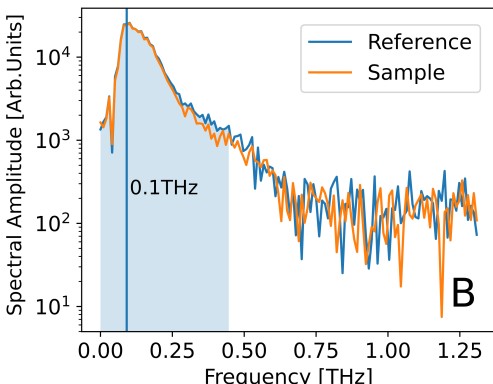

**Figure 4.** *Cont.*

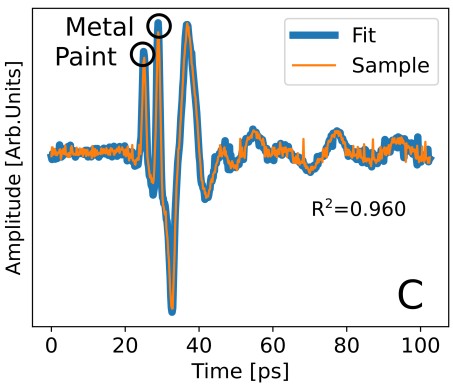
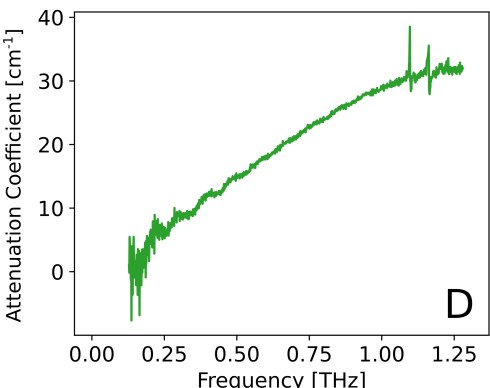

**Figure 4.** (**A**) Measurement of sample and a reference measurement at $\theta$ = 20 and s-polarization. The reference is a measurement of the backside of the paint sample. The circles mark the arrival time of each reflection. (**B**) FFT of (**A**). The shaded blue area contains approx. 90% of the signal power. The vertical blue line represents the maximum power output at 0.1 THz. (**C**) Fit using the two measurements from figure (**A**), results in a paint thickness of 232.8 µm. (**D**) Attenuation coefficient of silicone in the THz frequency domain, measured in a transmission setup.

### 3.2. Attenuation Coefficient of Silicone and Measuring Underwater

The silicone patch required for underwater operation introduced additional complications of signal attenuation by THz absorption in the silicone. The frequency-dependent attenuation coefficient for our silicone is shown in Figure 4D and was recorded using a transmission setup. The spectrum showed an increasing absorption towards higher frequencies. For a given output power of any THz system, it was therefore optimal to have a high power spectral density at low THz frequencies to maximize the total power arriving at the receiver. Figure 4B shows that the THz-CCS system had a maximum output at 0.1 THz and that 90% of the total electric field resided below 0.4 THz. In this regime, the attenuation coefficient was $\leq 10\,\mathrm{cm}^{-1}$, which for comparison was 20 times lower than for water [24]. To determine the efficiency of the silicone contact patch, two measurements through the contact patch onto a metal surface in dry conditions and submerged in water in Figure 5A could be considered, where there seemed to be no difference between the two measurements. Under the assumption that all materials did not have a magnetic response to the incident THz field, then the square of the electric field would be proportional to the intensity. From this, we could compute a measure $I(t)$ which was proportional to the energy of the reflected signal by summing over the squared electric field amplitudes $A(t)$. This could be used to calculate a measure of the total energy loss between a dry and submerged environment:

$$\text{Energy loss} = 1 - \frac{I_{\text{water}}(t)}{I_{\text{dry}}(t)} = 1 - \frac{\sum_t A_{\text{water}}(t)^2}{\sum_t A_{\text{ref}}(t)^2} = 1 - 98\% = 2\% \tag{9}$$

This resulted in a 2% energy loss, which could be a result of a thin layer of water between the silicone and the sample. An energy loss of 2% translated to a water layer between the contact patch and the sample of 0.5 µm, as the beams traveled through the layer twice in the reflection setup.

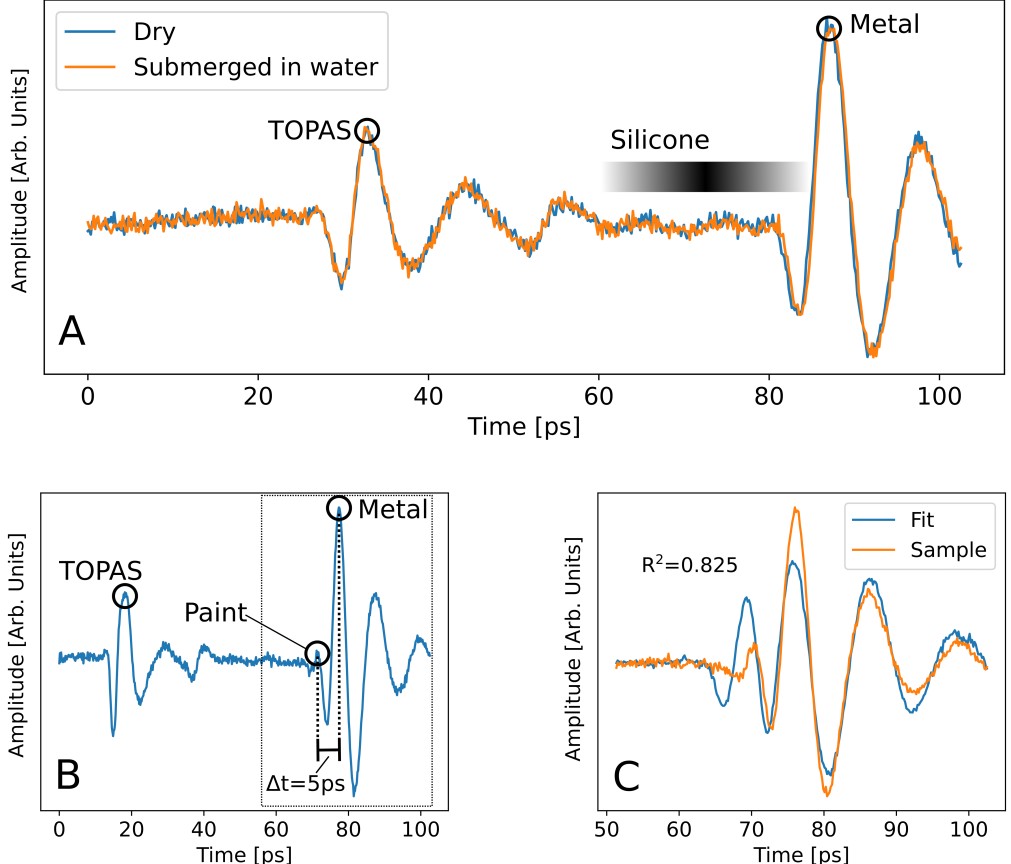

**Figure 5.** (**A**) Measurements on a metal surface through the developed contact patch in dry and submerged conditions. The fading bar shows the expected location of the silicone reflection. (**B**) Measurement of the sample through the developed contact patch underwater. Notice that there is no reflection at the plastic/silicone interface. (**C**) Fit on the measurement from (**B**) when only considering the signal from 50 ps ($R^2$ score of 0.825).

### 3.3. Measurement of Paint Sample through Contact Patch Underwater and Application of Fitting Algorithm

The measurement through the contact patch underwater of the paint sample can be seen in Figure 5B. The signal is visually different from the measurement in Figure 5A, where only a metal surface was measured. The time delay between the two rightmost marked peaks in Figure 5B was 5 ps, which, combined with the additional refraction of light through the TOPAS plastic, resulted in a thickness of 317 μm using Equation (1). This illustrated the result of insufficient visual determination of two reflection peaks.

An additional observation is the lack of reflection between the TOPAS plastic and the silicone contact patch, as expected due to the similar refractive indices of the two layers. This justifies the use of the simplified physical model shown in Figure 2E. Finally, the fitting algorithm was applied and a paint thickness was determined to be $232.8^{+6.3}_{-3.1}$ μm as listed in Table 1. The best fit, with a coefficient of determination of 0.825, can be seen in Figure 5C. The calculated paint thickness deviated by 1.2% from the measured thickness by the Alicona 3D scan. The high accuracy on the thickness determination despite the low $R^2$-value was due to the fact that the thickness estimation was robust to differences in the fitted and measured pulse shapes, since the temporal distance between reflections was proportional to $n \times d$, which was independent of the specific pulse shape. It was therefore possible to attain accurate and precise thickness measurements even though the correct pulse shape was not perfectly reconstructed, which tended to decrease the $R^2$-value. The difference between the fit and the measurement could be due to the assumption of a constant index of refraction for the TOPAS and silicone layers. A better fit could likely be produced by

using a more complex model for the index of refraction of the absorbing silicone such as the previously mentioned Debye model, as compared to using a constant index of refraction. This was not done due to convergence issues, but it is a topic of future work.

**Table 1.** The Alicona 3D thickness denotes the average and standard deviation of DFTs from the cross-sectional point measurements. The THz-CCS measurements were single-point measurements with 68% confidence limits.

| Type of Measurement | Thickness (μm) |
|---|---|
| Estimated DFT based on the amount of wet paint applied | 200 |
| Average DFT from Alicona 3D scan | $230.0 \pm 9.4$ |
| DFT from THz-CCS measurement without water | $235.0 \pm 0.2$ |
| DFT from THz-CCS measurement in water | $232.8^{+6.3}_{-3.1}$ |

## 4. Conclusions

This study demonstrated the concept of using THz-CCS to measure paint thickness underwater. A 3D height profile scan of a paint sample revealed that the average thickness of the paint sample across its width was 230.0 μm. To displace water, a silicone contact patch was created by 3D-printed molds, and it proved to have the correct firmness and an acceptable attenuation coefficient. The silicone contact patch was used in combination with a TOPAS plastic disc to create a water-displacing mechanism. The contact patch proved to be able to displace practically all water. The THz radiation was guided through the TOPAS plastic and the contact patch to interact with the paint and ultimately carry paint information back to a receiver in a reflection setup. Finally, a spectroscopic model was used in comparison to a simple time-of-flight analysis to determine the thickness of a maritime paint layer. The best fit of the spectroscopic model was able to determine the single-layer thickness of paint submerged in water with 1.2% deviation from the 3D height profile scan.

We expect our results to pave the way for the development of a novel THz-CCS-based device that can monitor and help maintain antifouling layers on ships. This will in turn help the shipping industry reach the emission reduction goals that they have set forward.

**Author Contributions:** J.H.-N.: writing, data analysis, investigation, and visualizations. J.Ø.K.: writing, data analysis, investigation, and visualizations. T.B.: supervision, conceptualization, and fundraising. B.H.M.: software and algorithm development, writing, review and editing. S.J.L.: supervision, conceptualization, review and editing, and project administration. P.U.J.: review and editing. All authors have read and agreed to the published version of the manuscript.

**Funding:** This research has been funded by The Danish Maritime Foundation and DTU Electro.

**Institutional Review Board Statement:** Not applicable.

**Data Availability Statement:** Data used in this study will be available upon request.

**Acknowledgments:** This project was solely possible due to having access to the THz-CCS system designed and provided by DTU Electro, as well as the paint samples provided by Jotun.

**Conflicts of Interest:** The authors declare no conflict of interest.

## Abbreviations

The following abbreviations are used in this manuscript:

| | |
|---|---|
| THz | Terahertz |
| CCS | Cross-correlation spectroscopy |
| FFT | Fast Fourier transform |
| DFT | Dry film thickness |

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
