# Peer review of "Measuring Maritime Paint Thickness under Water Using THz Cross-Correlation Spectroscopy"

_applsci, doi:10.3390/app122312397_

Round 1

Reviewer 1 Report

The submitted paper “Measuring maritime paint thickness under water using THz cross-correlation spectroscopy” by J.O. Knarreborg, J. Hjortshoj-Nielsen, B. H. Molvig, T. Bak, P.U. Jepsen and S. J. Lange is devoted to development of THz system for testing of paint thickness on ship hulls underwater. Applications of THz-TDS for measurement of paint thickness on various industrial and art objects were demonstrated many years before. Nevertheless the underwater applications can be of interest. The authors propose the ways for solving technical problems of making terahertz probe “optical contact” to the test object underwater as well as of THz-TDS data analysis for small time delays between THz waveforms, multiple reflection model and proper numerical optimization procedure. The results can have practical impact and look original and relevant. The paper is well written (the only small issue is that in caption to Fig. 2 the E panel is not described). I think this paper can be accepted for publication in Applied Sciences in its present form.

Author Response

Dear Reviewer, thank you for your feedback. The E panel now has a description, thank you for pointing this out. We are glad to hear that you found the article able to have a practical impact.

Reviewer 2 Report

This manuscript reports a THz system for the measurement of a maritime paint thickness under water. The authors present the technical realization and the data analysis as well as the practical demonstration on a real sample. 

Overall, I enjoyed reading the manuscript. The text is written very clearly. The motivation of the work, possible difficulties, alternative approaches and other aspects are described in the introductory part. This is followed by the measurement routine, the fitting and the analysis of the measurement precision. The presentation seems to be comprehensive. The only thing that I missed is the error for the thickness estimation from the THz-CCS in Table 1. The 3D microscopy estimation has the error of 9.4 um, but the THz don't have any. Or it was meant as the variation of the thickness across the sample's surface in the case of microscopy? Still, for the THz-CCS there should be a confidence interval for the estimated thickness resulting from the fitting procedure. It would useful to present it.

Author Response

Dear Reviewer, thank you for your feedback. We are very glad that you enjoyed the article. You are correct that table 1 had some shortages and ambiguity. We have added the confidence intervals for the two THz-CCS measurements, while also clearly stating that the 9.4 um is the standard deviation of the thickness across the sample surface. Thank you for your feedback once again.

Reviewer 3 Report

Please see the attached PDF

Author Response

Esteemed Sir/Madam,

We believe that the current manuscript presents new and important additions to the community that have not been properly weighed in your feedback. We also believe that you have misunderstood some important aspects of our work, which we seek to clarify here.

It is generally thought of in the community that THz is not compatible with transmission through water. In this work, we present an innovative way to deal with this problem in order to apply THz technology in water. It is of course well known that THz technology can measure layer thicknesses, but the demonstration of this technique under water is, together with the presented analysis method, novel.

The solution utilizes a stack of 2 materials with the same refractive index to make them look like 1 optical material. One is solid to provide mechanical stability while the other is deformable to ensure optimal contact and thereby push water out of the beam path. The used materials are TOPAS and silicone. We understand from your comments that you have understood that we use silicon, which we do not. The choice of index matching allows use to optimize the transmitted THz signal to the sample and to model the reflected signal in a simpler and more robust way.

Our choice of THz-TDS system is a new THz-CCS system, which is not a widely used and accepted system architecture in the community. In this work, we demonstrate for the first time that our system is capable of performing correct thickness measurements. This result is important and in our opinion highly relevant to the community, since new THz system architectures are constantly being developed and compared in order to drive THz innovative solutions forward. This is also why we are convinced that the present work is suitable for Applied Sciences.

Our previous work on paper thickness demonstrated that our THz-CCS system was able to retrieve useful THz signal from paper. At that point, it was clear that more sophisticated data processing schemes were necessary to measure the thickness of a thin layer. The present manuscript demonstrates how we have successfully used such processing scheme as part of a solution to a very important real-life problem. This solution would be of less value to the community had it not been obtained with our system, since the final field application requires a robust and easily portable system. Essentially, we are bringing the community an important step closer to the application of THz in the real world with our current work.

Round 2

Reviewer 3 Report

Authors effort for further clarification is much appreciated.

I agree that using THz waves under water is generally not convenient because of their strong absorption. While not trying to undermine authors effort regarding the packaging/sealing of the THz setup, I am still not convinced that the content of this manuscript justifies a new publication. 

I would like to reinstate that this manuscript reads well and seems technically sound. However my concern about its take away message is still in place.